# On the Resin Transfer Molding (RTM) Infiltration of Fiber-Reinforced Composites Made by Tailored Fiber Placement

**DOI:** 10.3390/polym14224873

**Published:** 2022-11-12

**Authors:** Lars Bittrich, Julian Seuffert, Sarah Dietrich, Kai Uhlig, Tales de Vargas Lisboa, Luise Kärger, Axel Spickenheuer

**Affiliations:** 1Leibniz-Institut für Polymerforschung Dresden e. V. (IPF), 01069 Dresden, Germany; 2Institute of Vehicle Systems Technology (FAST), Karlsruhe Institute of Technology (KIT), 76131 Karlsruhe, Germany

**Keywords:** permeability, RTM, tailored fiber placement, micrographs

## Abstract

Tailored fiber placement (TFP) is a preform manufacturing process in which rovings made of fibrous material are stitched onto a base material, increasing the freedom for the placement of fibers. Due to the particular kinematics of the process, the infiltration of TFP preforms with resin transfer molding (RTM) is sensitive to multiple processes and material parameters, such as injection pressure, resin viscosity, and fiber architecture. An experimental study is conducted to investigate the influence of TFP manufacturing parameters on the infiltration process. A transparent RTM tool that enables visual tracking of the resin flow front was developed and constructed. Microsection evaluations were produced to observe the thickness of each part of the composite and evaluate the fiber volume content of that part. Qualitative results have shown that the infiltration process in TFP structures is strongly influenced by a top and bottom flow layer. The stitching points and the yarn also create channels for the resin to flow. Furthermore, the stitching creates some eye-like regions, which are resin-rich zones and are normally not taken into account during the infusion of TFP parts.

## 1. Introduction

Recently, the demand for energy-efficient systems has leveraged the use of fiber-reinforced plastics (FRP) lightweight composites in structural components [1,2]. These materials are increasingly being employed in aeronautical, aerospace, and automotive applications. By employing a variable-axial (VA) fiber design, stiffness and strength properties may be improved when compared to classical FRP designs [3,4,5,6]. Thereby, the term VA means varying the fiber orientation at the ply level. The desired performance of FRP composites is achieved by guiding the loads almost exclusively along the fiber orientation and thus minimizing the shear load of the matrix. For a technical realization, TFP technology, which was developed at Leibniz-Institut für Polymerforschung Dresden (Germany), is well suited. The basics and some applications of TFP technology are described in [7,8]. Generally, the placement of carbon fibers is usually carried out by stitching dry rovings, as shown in Figure 1. The roving is guided through a rotatable roving pipe onto a base material, where a sewing thread applied in the zig-zag pattern holds it in place [3].

Generally, the dry preforms manufactured by TFP are consolidated by liquid composite molding (LCM) [9], where the liquid resin is infused/injected into a mold cavity. The impregnation of the preforms is made either via resin transfer molding (RTM) or via liquid resin infusion (LRI) with pressures higher than 1 bar or just using vacuum, respectively [1,9]. The understanding of the infusion characteristics, in a broad and/or in a particular sense, is of great interest to the scientific community. First Pillai [10] and later Michaud [11] have presented an extensive review of unsaturated flow in LCM and its characteristics. Furthermore, research has been performed in different ways, such as experimental trial and error methods for numerical modeling of the process [12,13,14].

For both experimental tests and numerical simulations, permeability plays an important role in the infusion characteristics. The resin flow is normally described by Darcy’s law [12,15,16], which evaluates a flow inside of a porous media, and it depends on the gradient pressure ∇p, resin viscosity η, and the permeability *K* of the component, being translated as a volume-averaged fluid velocity $\tilde{v}$ [13,17].
(1)$$\tilde{v}=-\frac{K}{\eta }\nabla p\;\;.$$

Experimental work has been performed in order to obtain the flow front and, through Darcy’s law, to determine the permeability. In order to monitor the resin flow front (and the degree of cure), Rubino et al. [18] have used dielectric sensors aiming at evaluating microwave heating to improve the infusion’s filling time. Optical methods were also used. Hancioglu, Sozer, and Advani [14] have compared RTM and vacuum-assisted RTM (VARTM) by using “effective” permeability on the last so as to avoid numerical/experimental determinations of the compaction of the preform due to the vacuum. The authors have used a digital camera to track the resin front. Simulations were in good agreement with the experiments. Kuentzer et al. [19] have also considered the visual characterization of the front in the determination of the bulk permeability. In ref. [12], permeability experiments of textiles in several different institutions were produced. Parameters such as fluid, type of flow, and measuring strategy were varied. An important result of this collaborative work is that the measurement of permeability is still a complex task, with several error sources caused mainly by human sources.

(Semi)-analytical approaches and numerical simulations were also performed so as to determine the flow characteristics [14,19]. Carlone et al. [16] present a numerical multi-scale approach so as to solve the dual-scale flow by first evaluating the bulk permeability and the saturation at the mesoscale level and, secondly, inputting these results into a macroscale model. Good agreement was found in numerical and experimental comparisons. Rubino and Carlone [20] have developed a semi-analytical model to determine the filling time of RTM and VARTM processes. The inclusion of the preform compression is required due to the compaction in the VARTM. It has been found that the larger the preforms’ compliance, the longer the filling time is.

TFP preforms have some particular characteristics that might influence the permeability and, thus, the infusion. Uhlig et al. [8] have experimentally evaluated the influence of several TFP parameters on unidirectional FRP (UD-FRP) coupons and found that they play a major role in the fiber volume content of TFP layers and rovings as well as the roving waviness. Particularly, the stitch width has a strong effect on the layer and rovings fiber contents. Seuffert et al. [13] have simulated the parallel and transverse to fiber intra-bundle permeability of TFP-made specimens. As stated in [10,11,19], woven and stitched fabric has a dual-scale flow, bulk- and micro-flow, and they exhibit distinct impregnation rates, and since TFP has particular characteristics regarding the stitching patterns, the bulk flow might be influenced by them. In [21], the influence of stitching on the infiltration in thickness direction is investigated, and it was found that the thread significantly increases the permeability.

As a result, the influences of the stitching patterns and TFP parameters on the impregnation of TFP preforms are investigated. It is highlighted that infiltration evaluations have been the subject of several studies, as described above. However, also stated by some of these publications, the preforms’ structure has a strong influence on the infusion characteristics. To the authors’ knowledge, no work that deals with the infusion of TFP parts has been found and, as will be presented, some particular features of the manufacturing process play an important role in the infiltration outcomes. Normally, in complex preform manufacturing, the stitching pattern exhibits randomized correlations as rovings next to each other may have different lengths. Nonetheless, in TFP UD stitching, patterns are often symmetric or antisymmetric, and these examples offer extremes of the stitching influence on the infiltration process. A specific tool that optically measures the flow front, similar to [14,19], is developed, and tests with varying stitching parameters are conducted. The permeability in fiber and perpendicular to fiber direction are obtained through a modification of Darcy’s law that allows variation of the pressure during the infusion. Microanalyses were also produced to evaluate the characteristics of the specimens as well as to derive the layered structure of TFP parts.

The paper is divided into six sections. Section 2 introduces the experimental tool and specimen characteristics, while Section 3 presents the permeability definition and evaluation. Section 4 introduces the microsection analyses of the TFP structure. The results of the paper are presented and discussed in Section 5 of the paper. The conclusions can be found in Section 6.

## 2. Experimental Evaluation

### 2.1. RTM Tool and Measuring System

In order to evaluate the infusion and the permeability of TFP preforms, a special RTM tool was manufactured, and its main characteristics are shown in Figure 2. It consists of a metallic frame on which the cavity forming plate is positioned and covered by a glass plate, enabling the resin flow front to be tracked optically (see Figure 2a), similar to the approaches of [14,19]. To prevent the glass plate from bending due to the pressure, a stiffener is added on top of the glass plate. A quick-release clamping system is used for the handy opening and closing of the tool (see the red handlers in Figure 2 and Figure 3). The setup of Figure 2a enables different studies of infiltration by only changing the cavity plate (metallic tool). In this case, two rectangular cavities—350 × 75 × 1 (mm)—were used to perform two infusion tests simultaneously. The preforms are positioned inside these cavities with the aid of silicon profiles at their edges to prevent race-tracking. Furthermore, each cavity also has an o-ring to ensure the sealing. A pressurized resin reservoir is used to inject the resin into the cavities without the assistance of vacuum at the outlet (see Figure 2b). A pressure of 1 to 3 bar above atmospheric pressure is kept in the reservoir. Moreover, pressure sensors are used near the inlet to measure the pressure inside the cavity.

The resin flow tracking scheme is shown in Figure 3. At the RTM tool inlet, the pressurized resin holder is connected. On the opposite side, an outlet vent is attached. The solid red arrow shows the direction of the resin flow. The resin front assessment is made optically by a standard DSLR camera. Reference points engraved on the glass plate were used for optical distortion and perspective corrections. The acquisition times for each sensor were different: one photo every second and five assessed points on the pressure sensor. Moreover, these assessments were not synchronous.

### 2.2. Specimen Properties and Tests

Some of the intrinsic parameters of the TFP process are examined, and they are depicted in Figure 4. There, two rovings are presented side-by-side where the thick black solid line describes their intended path. The thin red solid lines correspond to the sewing thread in a zig-zag stitch, and symmetric stitching is observable—refer to Figure 2b for the antisymmetric stitching. The red dots represent the stitching points, and the horizontal and vertical distances between two sequential points are important parameters for the TFP technology. By the R-value (vertical distance), one controls the thickness, smoothness of the thickness variation, etc., whereas, by the distance of stitch (horizontal distance), one can define the waviness (which is a correlation between the R-value, the distance of the stitch, and the width of the roving). From the distance between rovings, the thickness is derived (see ref. [3] for further details).

In considering the parameters of Figure 4, 8 × 2 infiltrations were conducted. The specimens have the same dimensions of the cavity (350 × 75 × 1 (mm)) and are stitched in a UD fashion. The base material consisted of a glass woven fabric with 108 g/m${}^{2}$. Furthermore, the roving used was a Tenax^TM^ HTS 12K (800 tex), and the sewing thread was made of polyester (10 tex). The description of these specimens are presented in Table 1. As observed, the experiments were made in the fiber and in the perpendicular direction with symmetric and antisymmetric stitching patterns. The first group—experiments no. 01 and 02—is considered the reference for the infusions. In the second group—experiments no. 03 and 04—the R-value is reduced. The stitch distance is modified in the third group (experiments no. 05 and 06), while in the last one, the distance and R-value are varied, and consequently, the fiber volume content (FVC) $V_{f}$. Uhlig et al. [8] have studied some of these parameters and their influence on the fiber volume content, finding a strong influence on the stitching width. It is then expected to obtain the sensitivity of these parameters against the infusion properties such as the permeability and thickness. Microsection analyses of the specimens were also produced so that the characteristics of the infiltration could be evaluated in a more detailed fashion.

Thus, by considering only this naming setup, all the important information regarding the specimen is readily available. Furthermore, the room temperature was kept between 20 and 23 ${}^{\circ }$C. The epoxy resin used is L20 with EpH161 as the hardener, with a viscosity at room temperature of around 1000 mPa s. This combination generates a low-viscosity resin that cures at room temperature, has a processing time of around 90 min, and displays good impregnation with respect to most reinforcing fibers. Furthermore, the density, tensile modulus, and strength of the L20+EpH161 are 1158 kg/m${}^{3}$, 3.4 GPa, and 70.2 MPa, respectively [22].

## 3. Permeability Definition

The flow front required for the permeability determination was measured at discrete time steps by image series, in which the front determination is performed by manual recognition in each image. The flow front represents the averaged progress as a straight line [12] (or the average volume of the resin inside the chamber of a particular time), as shown in Figure 5: the red line in both figures represents the actual flow front while the straight black line at the bottom figure represents the averaged value.

Software was developed to deal with corrections/distortions from the acquisition system and was based on OpenCV [23]. The tolerance was found to be around 0.1 mm by comparing image data to known tool sizes. The flow is then defined as the average volume of resin inside the infusion chamber at a particular time.

During the experiments, the pressure remained mostly constant. Intentional and unintentional changes in pressure are corrected by accounting for a modified Darcy’s Law and trapezoid integration. The permeability can be defined by Darcy’s law as
(2)$$\frac{dx_{f}}{dt}=-\frac{K\;\Delta p(t)}{\left(1-V_{f}\right)\;\eta \;x_{f}\left(t\right)}\;\;,$$
where $x_{f}$ defines the position of the flow front (see Figure 5), *K* corresponds to the permeability, Δp determines the difference of pressure between the inlet and the outlet, and η is the resin’s viscosity. Here a non-constant pressure is explicitly considered to account for slight changes in pressure at the beginning of the experiment. Equation (Equation 2) can be solved as
(3)$$x_{f}^{2}\left(t\right)=\frac{2\;K}{(1-V_{f})\;\eta }\int_{0}^{t}\Delta p\left(t\right)dt\;\;.$$

From Equation (Equation 2) to (Equation 3), two hypotheses are considered: constant viscosity and constant permeability over the entire infusion. The epoxy resin used in the infiltrations—L20 with Ep161-hardener—has little variation in viscosity under room temperature for around 30 min. The infiltration tests took around 15 min to be finished. The constant permeability is also supported since the flow media cross-section is the same in the specimen.

The viscosity, the fiber volume content, and the difference in pressure and its variances are known a priori. Equation (Equation 3) is solved by the trapezoidal rule, and the permeability is derived as follows
(4)$$K_{k}=\frac{x_{k}^{2}\;\left(1-V_{f}\right)\;\eta }{\sum_{j=1}^{k}\left(p_{j}+p_{j-1}\right)\left(t_{j}-t_{j-1}\right)}$$
in which the indices *j* and *k* describe time-based measurement positions for both the pressure and the resin flow position. Following Equation (Equation 4), the permeability can be defined even with variations in pressure. The time synchronization issue required for connecting the pressure data with the measurements of the flow front is solved by assuming constant permeability over time and changes in pressure.

## 4. Microsection Analysis

After the infiltrations and aiming at evaluating the FVC of the specimens in different regions of the cross-section, microsection analyses were performed in different positions of the specimens. First, a stitching scheme is introduced in Figure 6 and represents the symmetric fashion. It is important to highlight that the stitching points go through the roving in both symmetric and antisymmetric cases; however, in the latter, the two stitching points are close together. This information can be verified in Figure 7, in which the cross-sections of the two specimens are shown, where at the top and bottom, the symmetric and antisymmetric patterns are observed, respectively. The red dots represent the stitching yarn. Another important characteristic observed in Figure 6 is the eye-like structure developed by the stitching point. Due to the bending stiffness of the fibers belonging to the roving, the needle opens such a region by piercing through the roving. The importance of this eye-like structure for the infusion is essentially twofold: it creates a resin-rich zone, and it eases the resin flow from bottom to top or vice-versa. These channels behave differently regarding the stitching pattern—in the antisymmetric one, they are connected. This statement can be observed in Figure 7, where a comparison between the microsection of each stitching pattern is performed.

In the microsection analyses, information regarding the “local” thicknesses was also obtained. Figure 8 shows the measurements of the part and subpart thicknesses. Furthermore, each microsection provides more than one value of the assessed parameters, then possibly obtaining some variance in these values. The measurements were also taken in two different regions of the specimen (one close to the inlet while the other close to the outlet). By producing such microanalyses, it verified a layered pattern in the specimens: sewing thread layer, base material, roving layer, and sewing top layer. This aspect will be addressed in Section 5.

## 5. Results and Discussion

### 5.1. Thickness Variations Due to the Experimental Tool

Before describing the obtained results, it is important to highlight some problems in the tool and their effects. Dust particles between the metallic tool and the glass plate (see Figure 2) have probably caused some thickness variation in the specimens and also have led to serious damage to the glass plate. Furthermore, due to measures successfully taken to avoid race-tracking, such as silicon sealing, some other variations were observed. For a flow in the perpendicular direction to the preform, the race-tracking was of almost no concern due to the cut edges, preventing the resin from flowing in the gap between the preform and cavity edge. However, small deviations in the length of the cut fibers lead to a compression in the preform between the cavity edges, wrinkling them and influencing infiltration behavior. This problem does not occur for resin flow in the fiber direction since no cut is made in fibers in this direction. Moreover, small imperfections in the silicon sealing placement might have increased the deviation in the thickness. The preforms’ size has also created some problems: if the preforms were a little larger than the cavity, some waviness occurred due to side compression of the preform; if they were smaller, race-tracking was observable, even with the silicon sealing. As a consequence, the cavity depths during the experiments were larger than expected, and they varied from infusion to infusion and also between the specimens. Thus, the specimens’ thicknesses were not the same. However, qualitative results are herein evaluated, and they have presented interesting findings regarding the infiltration in TFP patches.

### 5.2. Permeability of TFP Preforms

Figure 9 shows the permeability and thickness of the specimens. The permeability error bars are defined by the variation during specimen infiltration, whereas their equivalent in thickness is derived from measurements at different positions in the specimen and inside a microsection, as mentioned above. It is important to highlight that the permeability was corrected with the measured averaged fiber volume content, $V_{f}$, and there is still a strong dependence on the permeability on the cavity size.

As a general trend, it is observable that the permeability in the fiber direction is slightly higher with an antisymmetric stitching pattern than with the symmetric one. The only case in which the symmetric permeability is higher than the antisymmetric one is experiment no. 7. It is also highlighted that, for all cases of infusion in the fiber direction, the antisymmetric specimens had larger thicknesses than the symmetric ones. Therefore, the outcomes of the permeability in the fiber direction are inconclusive: due to the cavity variation and the obtained data, one cannot define if there exist significant differences in the permeability regarding the stitching pattern.

For the perpendicular flow case, the permeability varies strongly from experiment to experiment. Experiment no. 2 has shown an opposite direction than experiment no. 4. Although all evaluations but no. 2 presented larger permeability with antisymmetric stitching than with symmetric one, little information can be extracted from these results due to the problems in the cavity size. Similarly, the thickness difference between the specimen inside the same group was not large when experiment no. 4 was excluded.

One possible reason for the slightly larger permeability in the case of antisymmetric stitching pattern, besides the larger cavity thickness, is that, essentially, the infusion of TFP preforms is dominated by the flow in-between the sewing thread layer (bottom) and the flow in the sewing top layer part of the preform (top). The antisymmetric pattern creates connections between the yarns, which are channels where the resin can flow easier. In the symmetric pattern, the channels are not connected to each other. These channels are fed and retrofed with resin by both top and bottom flows. This interaction accelerates the infusion process and might also be responsible for the low difference between the permeabilities in the fiber compared to the perpendicular to the fiber direction. Furthermore, as observable in Figure 6 and Figure 7, the sewing thread induces void regions throughout the preform, and, normally, these regions were not taken into account in the $V_{f}$ determination.

### 5.3. Thickness Distribution of the Specimens

As aforementioned, a layered structure is observed in the TFP specimens. Through the microsection analyses, the thickness distribution inside the specimens could be observed and measured. Figure 10 depicts the thicknesses of the sewing thread layer, base material, roving layer, and sewing top layer for each specimen. It is noticeable that the thickness of the sewing thread layer, base material, and roving layer vary only a little. The sewing thread layer and the base-material thicknesses vary between 0.111 and 0.133 mm and 0.108 and 0.121 mm, respectively. No connection between the overall specimen thickness with the small variations of these two thicknesses was found, indicating that these values are not influenced by the cavity size. Another important piece of information obtained by the microsection analyses is that the roving thickness also has little variation among the samples. Similarly to the sewing thread layer and the base-material thicknesses, no correspondence between the specimen thickness and the roving region was found. The only thickness that varies with the final cavity height is the thickness of the resin layer at the top of the specimen. The indication of these results is that the local fiber density remains high even for big cavity sizes. Furthermore, local $V_{f}$ is not strongly dependent on average $V_{f}$ as TFP preforms are compressed on their own. For the same reason, compression of the preforms through a small cavity is also limited since the sewing thread reduces the free movement of the fibers of the rovings. The implication, qualitatively, is that the local $V_{f}$ (the content in the roving region) has top and bottom limits. Further evaluation must be performed to obtain such limits and the influence of the stitching parameters on them.

## 6. Conclusions

This paper presents an evaluation of the infusion characteristics of TFP preforms. Despite the problems found with the cavity size, the volume content of the sewing thread layer, base material, and roving layer remained very similar in all experiments. This indicates that the local thickness of these regions is strongly dependent not on the cavity size but on the preform stitching, and, consequently, the $V_{f}$ of the roving region is strongly driven by the stitching pattern.

From the microanalyses, the following main observations are made: (i) the thread material creates empty spaces that are filled with resin but were always neglected in the definition of the fiber volume content, (ii) beneath the base material, there is a (thin) layer of resin, and (iii) eye-like structures are created around the stitching points, which are also filled with resin. The tendency shows that the permeability in the fiber’s direction is slightly higher with antisymmetric stitching compared to the symmetric one, which might result from the connected flow channels that the stitching creates. Still, the permeability variation results are inconclusive for both fiber and perpendicular to fiber direction with respect to the stitching pattern. Further investigations must be carried out to gather more evidence due to the cavity size variation and its influence on permeability determination. Little information can be taken from the permeability evaluated in the perpendicular direction.

It is highlighted that, due to the thickness variation of the specimens, this evaluation and its results must be viewed as a guideline in a qualitative way. Still, several tendencies could be captured by the experimental campaign, and the results work as a first step toward a guideline for the infusion of TFP patches and as a base for new experimental procedures. 

## Figures and Tables

**Figure 1 polymers-14-04873-f001:**
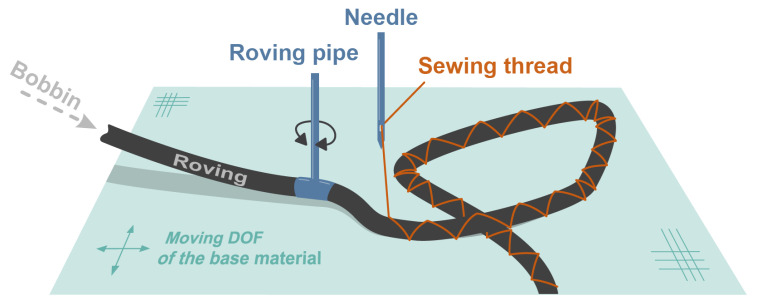
Basic principle of the TFP process [3].

**Figure 2 polymers-14-04873-f002:**
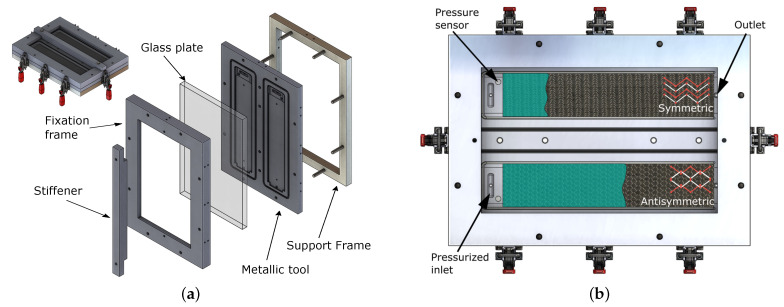
RTM tool, showing (**a**) the tool and its detailed exploded view and (**b**) the experimental idea and details of the metallic tool (with the cavities).

**Figure 3 polymers-14-04873-f003:**
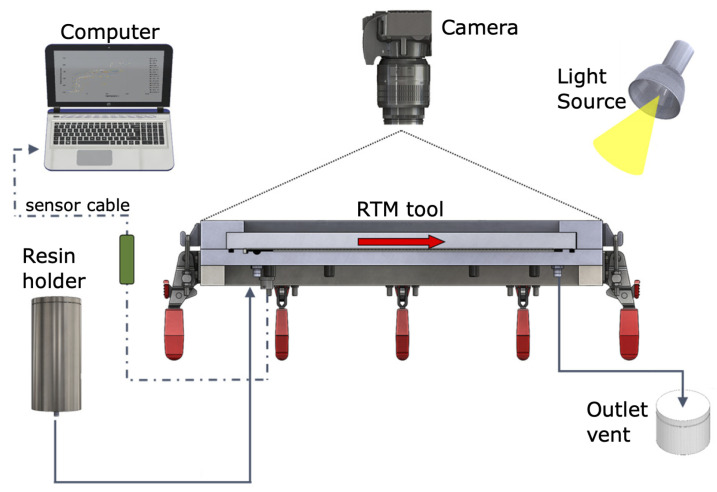
Experiment setup along with the measuring system.

**Figure 4 polymers-14-04873-f004:**
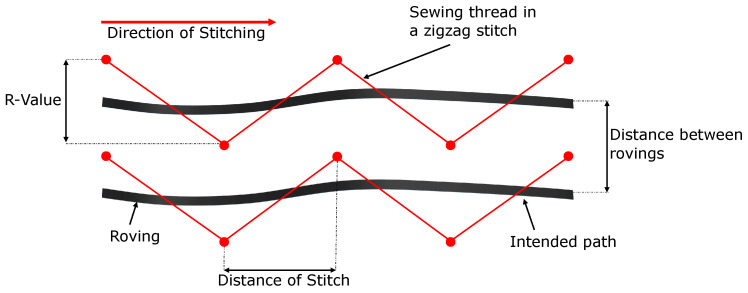
Stitching parameters where the black lines and red lines correspond to the rovings and the sewing thread, respectively.

**Figure 5 polymers-14-04873-f005:**
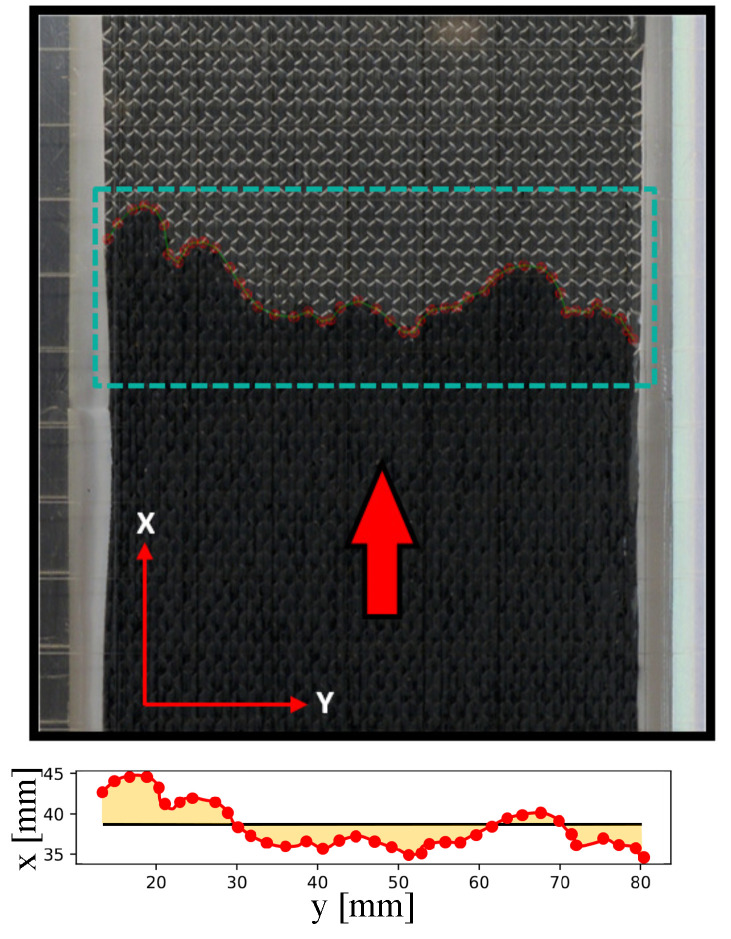
Flow front analysis example: the red arrow presents the flow direction (top figure), and the red line (bottom figure) is the manual recognition of the flow. In the bottom figure, the straight black line is the average value of the flow front.

**Figure 6 polymers-14-04873-f006:**
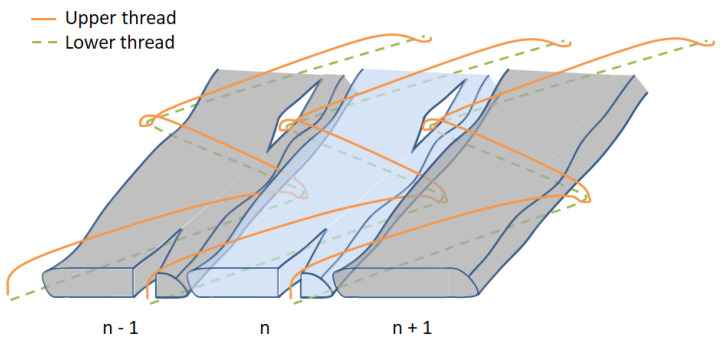
Scheme of the stitching (symmetric stitching).

**Figure 7 polymers-14-04873-f007:**
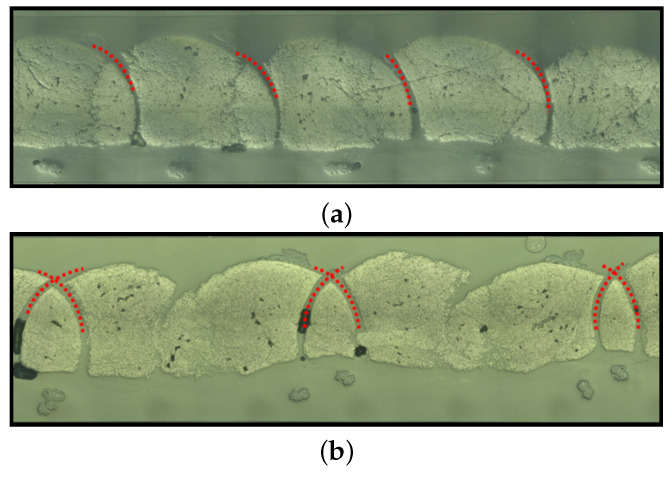
Micrography of the cross-section highlighting the (**a**) symmetric and (**b**) antisymmetric stitching.

**Figure 8 polymers-14-04873-f008:**
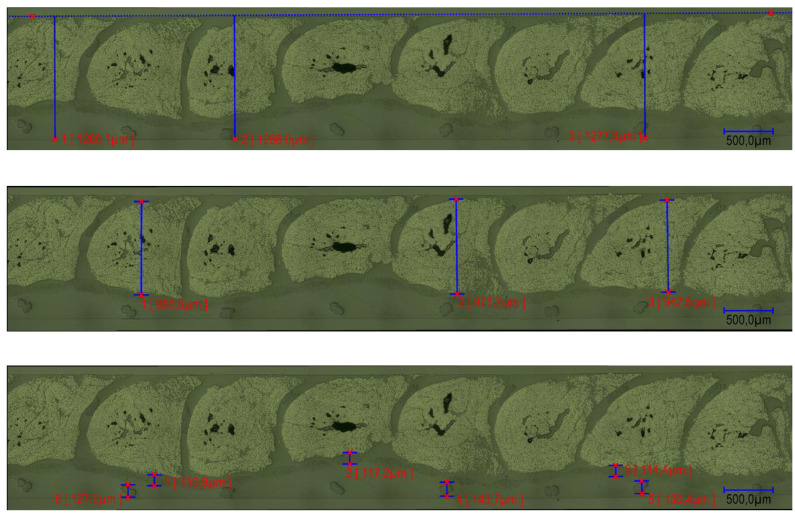
Examples of measurements of distance in specific regions at several places of a microsection: overall (**top**), roving (**middle**), and base-material and thread thicknesses (**bottom**).

**Figure 9 polymers-14-04873-f009:**
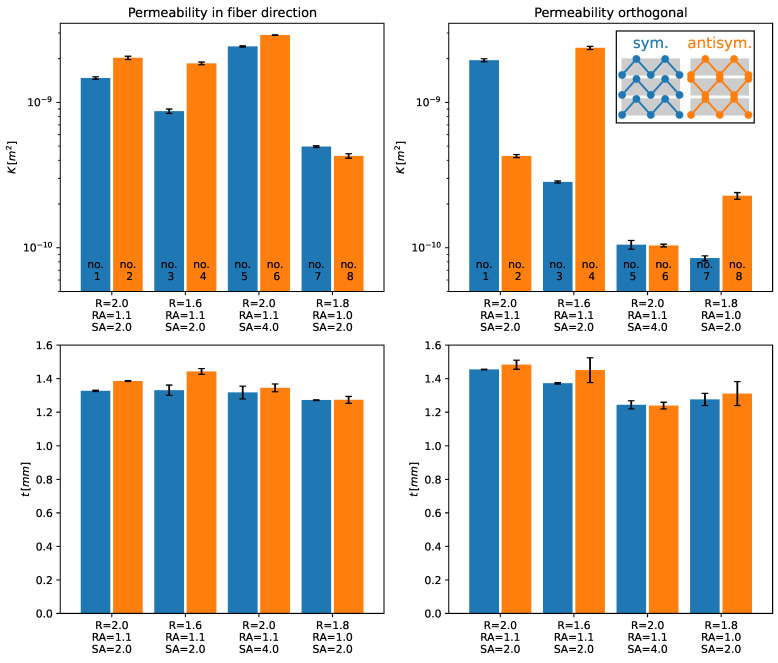
Permeability and thickness distribution of the specimens.

**Figure 10 polymers-14-04873-f010:**
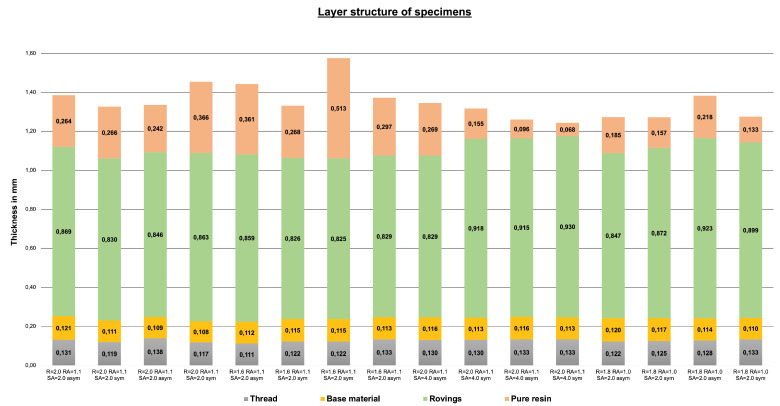
Thicknesses of sewing thread layer, base material, roving layer, and resin layer within each TFP specimen.

**Table 1 polymers-14-04873-t001:** Configuration of the different setups of the experimental campaign where the parameters in blue are varied with respect to the base setup.

Experiment Number	Stitching Type	Direction of Flow	Cavity	Distance (mm)	R-Value (mm)	Stitch *d*_min_ (mm)	*V_f_* (%)
01	symm.	Fiber	1	1.10	2.00	2.00	50
	asymm.	Fiber	2	1.10	2.00	2.00	50
02	symm.	Perpendicular	1	1.10	2.00	2.00	50
	asymm.	Perpendicular	2	1.10	2.00	2.00	50
03	symm.	Fiber	1	1.10	1.60	2.00	50
	asymm.	Fiber	2	1.10	1.60	2.00	50
04	symm.	Perpendicular	1	1.10	1.60	2.00	50
	asymm.	Perpendicular	2	1.10	1.60	2.00	50
05	symm.	Fiber	2	1.10	2.00	4.00	50
	asymm.	Fiber	1	1.10	2.00	4.00	50
06	symm.	Perpendicular	1	1.10	2.00	4.00	50
	asymm.	Perpendicular	2	1.10	2.00	4.00	50
07	symm.	Fiber	1	1.00	1.80	2.00	55
	asymm.	Fiber	2	1.00	1.80	2.00	55
08	symm.	Perpendicular	1	1.00	1.80	2.00	55
	asymm.	Perpendicular	2	1.00	1.80	2.00	55

## Data Availability

Not applicable.

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
