# Peer review of "On the Resin Transfer Molding (RTM) Infiltration of Fiber-Reinforced Composites Made by Tailored Fiber Placement"

_polymers, 2022, doi:10.3390/polym14224873_

Round 1

Reviewer 1 Report

The paper deals with a newly developed technique called as tailored fiber placement (TFP). The paper introduces methods/tools to evaluate resin infiltration, and also, presents results in terms of permeability and thickness for eight categories of specimens with varying values of stitch distance, R-value, and fiber volume content. Attention has been made to symmetric versus antisymmetric types of stitch as well as microsection analysis.

The paper does offer the novelty required for publication in the Journal, and the results/method presented by the paper are very useful and interesting for researchers and practitioners. The paper is very well-written with a logical structure and easy-to-follow. English is satisfactory. Thereby, the manuscript is recommended for publication. The following minor points are suggested to the authors for consideration:

1)      There are some abbreviations which were not introduced when appearing for the first time in the manuscript. These include: LCM (line 34), VARTM (line 62), UD-FRP (line 67). It is advised to identify them as what they stand for.

2)      Line 158: Although the type of resin and hardener are mentioned, it is however desirable to provide more quantitative details on their properties including mechanical characteristics.

3)       Line 168: A reference is needed for OpenCV.

4)      Lines 244-246: It is concluded that “the permeability in fiber direction is slightly higher with antisymmetric stitching pattern than the symmetric one.” And it is stated that the exception is experiment no. 7. Is there any reason for this exception or it is just an issue of test data variation? In other words, if the test is repeated, the same result is derived?

5)      Figure 10: The labels for specimens depicted under each figure are a bit confusing when compared to those in Table 1. For instance, experiments no. 7 and 8 have RA=1.1 according to Fig.1 whereas RA=1.00 in Table 1. Please clarify it or correct it if there is a typo. Similar comment goes for Figure 11.

6)       The TFP-made FRP can receive vast applications in structural engineering and seismic strengthening of infrastructures. From structural point of view, a very interesting application as a potential field of future research seems to be Near Surface mounted FRP reinforcement (NSM FRP). The following paper provides good insight to the NSM FRP technique from both experimental and numerical aspects, and thereby, it is suggested to the authors to include the following paper in the list of references:

Nasrollahzadeh, K.; Afzali, S. Fuzzy logic model for pullout capacity of near-surface-mounted FRP reinforcement bonded to concrete. Neural Computing and Applications 2019 Nov;31(11),7837-65. https://doi.org/10.1007/s00521-018-3590-2

Author Response

Please, see the document attached to this message.

Reviewer 2 Report

in this study, a series of experiments are conducted to investigate the influence of  TFP manufacturing parameters on the infiltration process. The experimental results are  presented well so that manuscript may be accepted without re-review .

Author Response

(The authors gave the same response as above.)
